# Exploring the Effectiveness of Diffusion Models in One-Shot Federated Learning

## Abstract

Federated learning (FL) enables multiple clients to train models collectively while preserving data privacy. However, FL faces challenges in terms of communication cost and data heterogeneity. One-shot federated learning has emerged as a solution by reducing communication rounds, improving efficiency, and providing better security against eavesdropping attacks. Nevertheless, data heterogeneity remains a significant challenge, impacting performance. This work explores the effectiveness of diffusion models in one-shot FL, demonstrating their applicability in addressing data heterogeneity and improving FL performance. Additionally, we investigate the utility of our diffusion model approach, FedDiff, compared to other one-shot FL methods under differential privacy (DP). Furthermore, to improve generated sample quality under DP settings, we propose a simple Fourier Magnitude Filtering (FMF) method, enhancing the effectiveness of the generated data for global model training. The code will be made publicly available.

## 1 Introduction

Federated learning (FL) is a distributed machine learning technique that enables multiple clients to participate in the training process in a privacy-preserving manner. In FL, each client trains a local model on its own data and sends the model to a central server. The server combines these updates to improve the global model, which is then sent back to the clients. This approach ensures that the clients' data is kept private while enabling the central server to learn from the collective knowledge of all participating users (Kairouz et al., 2019).

However, FL poses significant challenges in terms of communication cost and data heterogeneity across clients. Communication cost is a major bottleneck in FL systems, as clients need to communicate frequently with the server over multiple rounds during the training process (Kairouz et al., 2019; Sahu et al., 2018; Acar et al., 2021a). This leads to a high communication overhead, making the process slow or simply infeasible. To overcome this challenge, **one-shot federated learning** has recently gained traction in the research community (Guha et al., 2019; Zhang et al., 2022; Zhou et al., 2020; Salehkaleybar et al., 2021). In this setting, clients only communicate once with the server during the training process, significantly reducing the communication requirements. This approach not only improves the efficiency of the training process but also provides a better framework for privacy and application. Specifically, one-shot FL provides better security against eavesdropping attacks, where adversaries attempt to steal or tamper with the information being sent between clients and the server (Liu et al., 2022). By only requiring one round of communication, one-shot FL significantly reduces the likelihood of such attacks. Furthermore, traditional multi-round training may not be a practical option in some cases, such as that of model markets (Li et al., 2020). In these scenarios, models are trained to convergence by a participating user, and simply made available as a pretrained model to potential buyers, without any option for iterative communication.

However, another significant challenge in federated learning still remains, and that is, the data heterogeneity problem (Mendieta et al., 2021; Karimireddy et al., 2019; Li et al., 2021; Kairouz et al., 2019). In FL, clients often have very different data distributions, making optimization particularly challenging across the federated system. In the one-shot setting, this is especially detrimental to performance. Without the luxury of multiple communication rounds, the resulting models will be significantly biased towards their narrow data distribution and difficult to reconcile into a global model. Knowledge distillation-based approaches have been studied in the literature in an attempt to

address these problems (Guha et al., 2019; Li et al., 2020; Zhang et al., 2022). Nonetheless, these methods still struggle immensely under high heterogeneity, resulting in large drops in performance.

Yet, another class of model is potentially well-suited for capturing such narrow distributions at the client. Rather than reconciling discriminative models trained on the clients, one could instead leverage generative models, which can easily capture the narrow distributions present on clients in high heterogeneity scenarios. These generative models can then be gathered from the clients and inferenced on the server to form a dataset for global model training. (Heinbaugh et al., 2023) conducted a preliminary study of such a framework with conditional variational autoencoders (CVAEs) (Sohn et al., 2015) for one-shot FL, but there is still much to investigate in this paradigm. *Specifically, we consider two primary research questions (RQ) in this work.*

***RQ1***. First, we explore the utility of diffusion models in federated learning and their potential for improving the performance of the one-shot FL process. Diffusion models (Ho et al., 2020) have recently emerged as prominent approaches for image generation, inspiring our investigation. We suggest that specific traits of diffusion models could provide advantages for one-shot FL, as discussed in Section 3. We then validate this hypothesis through comprehensive experiments with our approach, FedDiff, across various settings.

***RQ2***. Second, we investigate one-shot FL methods under provable privacy budgets with differential privacy (DP), as this aspect is not directly addressed in most one-shot FL works. Safeguarding model privacy is critical in this setting, as the client models obtained in one-shot FL can be reused multiple times or even traded in a model market. Furthermore, in light of recent work (Carlini et al., 2023), we examine the potential memorization of diffusion models within our FedDiff approach and the effectiveness of DP as a mitigation strategy (Section 4).

After studying these research questions, we further explore a simple technique for improving the performance of our FedDiff method under DP settings. We observe that the quality of generated samples may deteriorate under DP constraints, rendering some samples counterproductive to the training of the global model. To improve the quality and consistency of the synthetic data, we propose a straightforward filtering approach, termed Fourier Magnitude Filtering (FMF). FMF leverages sample magnitudes derived from the Fourier transform to guide the selection of valuable samples. The resulting filtered dataset substantially improves the utility of the generated data, particularly in challenging conditions, as detailed in Section 4.2. Therefore, in this work, our **contributions** are summarized as follows:

- We contribute to the FL literature with the first study exploring diffusion models in one-shot federated learning. Our comprehensive investigation unveils the unique advantages inherent to diffusion models, which enhances the overall performance of one-shot FL challenges while also addressing the significant challenges of data heterogeneity. By employing diffusion models, we establish a novel approach, FedDiff, that not only ensures superior model performance but also aligns with the core requirements of one-shot FL.
- We further study the privacy and utility of both discriminative and generative-based SOTA one-shot FL methods with differential privacy (DP) guarantees under heterogeneous settings. We find that our FedDiff approach outperforms all other methods by a significant margin (from ∼5% to ∼20% across many datasets and settings), even when differential privacy is employed.
- While FedDiff performs very well, we note that sample quality is affected under DP. Therefore, to improve performance in such conditions, we propose a simple Fourier Magnitude Filtering (FMF) approach, which improves the effectiveness of the generated data for global model training by removing low-quality samples.

## 2 BACKGROUND AND PRELIMINARIES

### 2.1 ONE-SHOT FEDERATED LEARNING

Federated learning (FL) has emerged as a promising paradigm for collaborative machine learning across decentralized devices while preserving data privacy. The seminal work by McMahan et al. (McMahan et al., 2016) introduced the concept of FL, where model updates are computed locally on user devices and aggregated on a central server. However, in the standard FL process, many iterative communication rounds are required for convergence. One-shot FL, therefore, studies how

to effectively learn in this distributed setting in a single round, thereby mitigating the need for many communication rounds. Several approaches have been proposed to tackle the unique characteristics of one-shot FL. (Guha et al., 2019) introduce the one-shot federated learning framework and study several baseline approaches. In (Guha et al., 2019) and (Li et al., 2020), distillation approaches are studied using the ensemble of client models to the global model, and assume a public dataset for this purpose. However, such an assumption is limited, as public data related to the domain of interest is often not available. A data-free method within the distillation methodology was proposed by (Zhang et al., 2022), where a generative adversarial network (GAN) is trained at the server level to generate the data for distillation, and iteratively optimized between distilling to the server model and training the GAN with the ensemble of client models.

Nonetheless, these methods still struggle with heterogeneous environments, as we find in Section 3. Generative models on the client are well-fitted for better undertaking in such settings, as they can focus on the narrow client distributions and simply provide data at the central location. (Heinbaugh et al., 2023) introduce the use of CVAEs in highly heterogeneous one-shot FL. However, CVAEs exhibit suboptimal sample quality, a limitation that becomes markedly exacerbated with more complex datasets and when subjected to the constraints of DP, which are not explicitly addressed in the study by (Heinbaugh et al., 2023). In this work, we investigate diffusion models in one-shot FL and leverage their unique characteristics for the task, illustrating their potential in a variety of difficult FL settings and privacy guarantees.

## 2.2 DIFFUSION PROBABILISTIC MODELS

Diffusion probabilistic models (Ho et al., 2020; Dhariwal & Nichol, 2021), or simply diffusion models as they are now commonly referenced (DM), have gained traction for application in generative vision tasks. Simply put, DMs aim to learn the backward process that can iteratively denoise an image corrupted with Gaussian noise back to the original. Specifically, as detailed in (Ho et al., 2020), noise is introduced to a given sample via a Markovian chain forward process

$$q(x_{1:T}|x_0) = \prod_{t=1}^{T} q(x_t|x_{t-1}), \tag{1}$$

where $T$ is the total number of iterations (or timesteps) applied, and $q(x_t|x_{t-1})$ is parameterized by $\mathcal{N}(x_t; \sqrt{1 - \beta_t}x_{t-1}, \beta_t I)$. $\beta$ is a value between (0,1), and increases with timestep $t$, essentially making the final $q(x_T|x_0)$ approximately a simple Gaussian $\mathcal{N}(0, I)$. This forward process is fixed, and the goal of the diffusion model is to learn the reverse process. During training, we simply optimize for predicting the noise $\boldsymbol{\rho}$ from an arbitrary step $t$ in the forward process, forming a loss function (Ho et al., 2020)

$$L = \mathbb{E}_{t,\mathbf{x}_0,\boldsymbol{\rho}} \left[ \|\boldsymbol{\rho} - \boldsymbol{\rho}_\theta (x_t, t)\|^2 \right]. \tag{2}$$

The process can also be conditioned on another variable $y$ and giving $\boldsymbol{\rho}_\theta (x_t, y, t)$. For example, the diffusion model can be class conditioned (Ho & Salimans, 2022), with $y$ being a variable representing the class of the sample from a classification dataset. We utilize the class-conditioning approach of (Ho & Salimans, 2022) in our diffusion models for FL.

## 2.3 DIFFERENTIAL PRIVACY

Differential privacy (DP) (Dwork et al., 2006; 2014; Dwork, 2011) is a framework for ensuring that the output of a computation, such as machine learning model training, does not reveal sensitive information about any individual data point in the training dataset. A computation is said to be differentially private if the probability of obtaining a particular output is roughly the same whether a particular individual's data is included in the computation or not. Formally (Dwork et al., 2014),

$$Pr[A(D) \in S] \leq e^\epsilon \cdot Pr[A(D') \in S] + \delta, \tag{3}$$

where $A$ is a randomized algorithm, $D$ and $D'$ are a pair of datasets that differ in at most one record, and $S$ is any subset of the output space of $A$. $(\epsilon, \delta)$ control the level of privacy protection provided by the algorithm, essentially determining the maximum allowable amount of information that can be harnessed from the data. Larger values of $(\epsilon, \delta)$ correspond to weaker privacy guarantees, while smaller values of $(\epsilon, \delta)$ correspond to stronger guarantees.

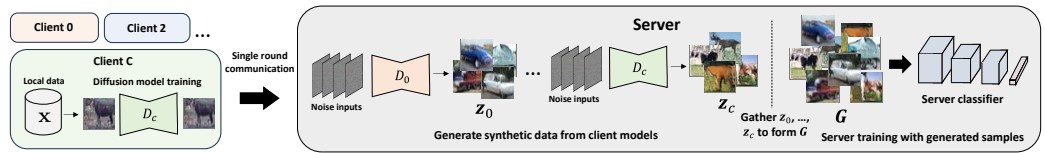

Figure 1: Our one-shot FL approach, FedDiff. We first train a class-conditioned diffusion model on local data $\mathbf{x}$ at the clients. After completing training, the local diffusion models $D_0, D_1, ..., D_c$ are gathered by the server, where they are used to generate data $\mathbf{z}_0, \mathbf{z}_0, ..., \mathbf{z}_c$, which are combined to form the global training data $\mathbf{G}$. The global model is then trained on this synthetic dataset $\mathbf{G}$.

To train deep learning models with such guarantees, differentially private stochastic gradient descent (DPSGD) is typically employed (Abadi et al., 2016). In DPSGD, two main mechanisms are used to protect the privacy of individual data points: per-sample gradient clipping and the addition of random noise to the clipped gradients. Per-sample gradient clipping involves setting a maximum threshold on the norm of the gradient computed for each data point, so that if the norm of a gradient exceeds the threshold, it is rescaled to have a norm equal to the threshold. This step is necessary to limit the sensitivity of the loss function, which measures how much the loss function changes when a single data point is removed from the training dataset. After the gradients have been clipped, random noise is added to them before they are used to update the model parameters. The amount of noise added is calibrated based on privacy budget parameters $(\epsilon, \delta)$.

## 3 DIFFUSION MODELS FOR FEDERATED LEARNING

Before delving into the underlying motivation for our research questions ***RQ1*** and ***RQ2***, it is essential to provide a brief exposition of the one-shot FL process when integrating generative models. The core premise of this approach departs from the traditional method of client-side discriminative model training. Instead, it advocates for the training of generative models on the client devices. These client-side generative models are aggregated and used offline on the server side to synthesize data, which, in turn, facilitates the training of a global discriminative model. Within the scope of our study, we undertake an investigation into the viability and efficacy of leveraging diffusion models within this paradigm.

**Why diffusion models?** In (Xiao et al., 2021), a generative learning trilemma is shown with model types, trading off sample quality, diversity, and fast sampling. CVAEs (as employed in (Heinbaugh et al., 2023)) are typically identified to excel in diversity and fast sampling, but lacking in sample quality. However, for one-shot federated learning, fast sampling is not a concern, as the sampling can be done offline at the server (Figure 1). Therefore, ***high sample quality and diversity are more valuable properties in one-shot FL***, as these will positively impact the performance of the trained global model with the synthetic data. In this trilemma, diffusion models excel in sample quality and diversity, but are not as quick to sample. This motivated us to investigate the potential of DMs in this setting, as the inherit strengths of DMs align with the needs of one-shot FL.

Furthermore, while CVAEs and diffusion models share a common origin in terms of their objective, they differ in their approach to achieving this objective. The optimization task of the diffusion model is simplified to learning a Markov process to reverse a fixed forward process. The training is structured such that the model only needs to learn how to denoise a small step in the generation process, breaking down the problem. In contrast, CVAEs must simultaneously learn both the forward process to encode the image to a latent space, and the decoding process from that latent vector. We reason that the simplified objective of DMs helps achieve superior performance when dealing with complex data within the challenging FL environment (data heterogeneity, class imbalance, and limited sample sizes). Moreover, in the FL setting, privacy is of critical importance. To ensure privacy, training is done with DP, which introduces noise to the training process and increases the difficulty of optimization. In these settings, the simpler training paradigm of diffusion models is particularly advantageous. We empirically investigate these motivations in the following sections.

### 3.1 EXPERIMENTAL SETUP

We begin by investigating ***RQ1*** with a study of diffusion models in one-shot FL. Our approach, FedDiff, is illustrated in Figure 1. We begin by training class-conditioned diffusion models using the local data $\mathbf{x}$ on the clients. After training, the server collects these local models, denoted as $D_0$,

$D_1$, ..., $D_c$, which are then used to generate data $\mathbf{z}_0$, $\mathbf{z}_1$, ..., $\mathbf{z}_c$. The label distributions from the clients are used to condition the generative models during generation, as in (Heinbaugh et al., 2023). The combination of these synthesized samples forms our global training dataset, $\mathbf{G}$. Subsequently, the global model is trained on the synthetic dataset $\mathbf{G}$ and evaluated in our experiments.

**Comparison Methods.** We compare with key baselines and the most recent state-of-the-art one-shot FL methods throughout our investigation.

*FedAvg* (McMahan et al., 2016) is a standard baseline, which simply trains discriminative classifiers at the clients and averages their parameters, typically weighted by the number of samples at each client, to form a single server model.

*DENSE* (Zhang et al., 2022) is a one-shot FL approach that first trains the discriminative classifiers on the clients to convergence. Once the client models are collected, it performs two stages of training in an interactive manner, switching between training a GAN-based network for generating synthetic data and using the synthetic data to distill the ensemble of client models to a single server model.

*OneShot-Ens*. We also include an idealized variant of DENSE, where rather than attempt to distill the ensemble of client models to a single server model, we simply employ the ensemble as the final model, as shown in (Guha et al., 2019) and similarly compared to in (Heinbaugh et al., 2023). We term this approach OneShot-Ens throughout the paper.

*FedCVAE* (Heinbaugh et al., 2023). This recently proposed method employs conditional variational autoencoders (CVAEs) for one-shot federated learning. Their approach has two variants, FedCVAE-KD and FedCVAE-Ens, which differ in how they operate at the server level. FedCVAE-KD distills all generative models from the clients to a single CVAE, and then generates data for training the global model. On the other hand, FedCVAE-Ens employs each client model to generate data, contributing to the final dataset for training the server model. The latter variant always shows significantly better performance than the other in their paper; therefore, we compare with this FedCVAE-Ens variant and refer to it as FedCVAE in the rest of the paper.

**Datasets.** We employ three datasets, FashionMNIST (Xiao et al., 2017), PathMNIST (Yang et al., 2023), and CIFAR-10 (Krizhevsky et al., 2009), which provide a range of domains and complexities. More details on the datasets are provided in Appendix A.1. For our experiments, we divide the training set among $C$ clients with a Dirichlet distribution $Dir(\alpha)$, as commonly done in FL literature (Mendieta et al., 2021; Acar et al., 2021b; He et al., 2020; Heinbaugh et al., 2023). This partitioning approach creates imbalanced subsets, where some clients may not have any samples for certain classes. As a result, a significant number of clients will only encounter a small subset (or potentially only one) of the available class instances. We visualize data distributions with $Dir(0.1)$ and $Dir(0.01)$ across 10 clients in Figure 5 of the Appendix.

**Federated Learning Settings.** We reproduce DENSE and FedCVAE for our settings with their respective official code repositories. For all experiments, we perform 3 independent runs with different seeds and report the mean and standard deviation. For all approaches, we train client models for 200 local epochs. For DENSE, FedCVAE, and FedDiff, we train the final global model for 50 epochs. The global model is the same for all methods, and consists of a 4-layer network with two convolutional layers and two fully connected layers, as in (Heinbaugh et al., 2023). For the generator of FedCVAE, we employ their CVAE variant with residual blocks, which has approximately 5.9M parameters. For our diffusion model, we employ a basic U-Net structure with residual blocks (Ho et al., 2020; Ronneberger et al., 2015) and class-conditioning, with similar parameters to FedC-VAE ($\sim$5.8M). For experiments with differential privacy, we employ the Opacus (Yousefpour et al., 2021) library in PyTorch (Paszke et al., 2017) to track privacy budgets. Further training details and additional experiments can be found in the Appendix A.1 and A.3. In the following sections, we explore many different data heterogeneity levels and the number of clients.

## 3.2 DATA HETEROGENEITY

Data heterogeneity is a critical challenge in FL, particularly with one-shot settings. Even in the standard FL scenario of multiple communication rounds, client models often fit to very different distributions, and effectively reconciling their learnings is daunting. This is exacerbated in the one-shot setting, as we no longer have the luxury of getting many iterations to progressively steer the learning process towards an ideal encompassing representation.

Table 1: Data heterogeneity results with various $Dir(\alpha)$ partitions. Smaller alpha values indicate higher levels of heterogeneity. Typical approaches leveraging discriminative models rapidly degrade in performance as heterogeneity increases. However, generative approaches are more robust to such conditions. **Our FedDiff shows superior performance to all, particularly in the most challenging scenarios (CIFAR-10, high heterogeneity).**

|  | $Dir(\alpha)$ | FedAvg | DENSE | OneShot-Ens | FedCVAE | **FedDiff** |
|---|---|---|---|---|---|---|
| FashionMNIST | $\alpha = 0.1$ | 56.24±3.05 | 64.77±2.77 | 66.81±1.62 | 75.70±1.87 | **87.21±0.74** |
|  | $\alpha = 0.01$ | 28.62±9.74 | 28.17±17.7 | 33.07±17.5 | 76.31±2.72 | **86.81±0.54** |
|  | $\alpha = 0.001$ | 25.00±5.33 | 27.36±3.97 | 31.10±4.09 | 79.05±0.21 | **86.59±0.69** |
| PathMNIST | $\alpha = 0.1$ | 23.91±3.89 | 48.09±3.25 | 30.20±4.37 | 41.94±0.52 | **74.58±1.02** |
|  | $\alpha = 0.01$ | 18.21±9.19 | 26.83±10.2 | 30.61±9.55 | 45.16±0.55 | **70.61±1.37** |
|  | $\alpha = 0.001$ | 17.99±4.67 | 25.19±4.87 | 29.84±5.32 | 47.63±2.83 | **69.43±1.30** |
| CIFAR-10 | $\alpha = 0.1$ | 14.27±3.24 | 34.96±8.16 | 37.83±6.91 | 31.84±0.63 | **57.69±2.07** |
|  | $\alpha = 0.01$ | 13.57±3.45 | 20.24±0.84 | 21.49±0.81 | 33.46±2.88 | **56.57±2.42** |
|  | $\alpha = 0.001$ | 12.51±4.81 | 10.77±2.52 | 18.53±9.49 | 34.40±2.06 | **55.75±1.55** |

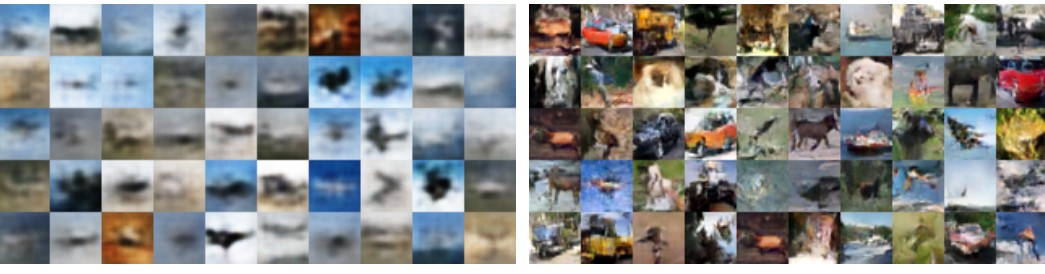

(a) FedCVAE                                      (b) FedDiff (ours)

Figure 2: Random sets of generated samples from FedCVAE and our FedDiff approach. By leveraging the intrinsic properties of diffusion models (DMs), which are well-aligned with the requirements of one-shot Federated Learning (FL), we achieve substantial benefits in sample quality and subsequent global model performance.

In Table 1, we analyze the performance of all methods under moderated ($Dir(0.1)$) to extreme ($Dir(0.001)$) heterogeneity. Interestingly, FedDiff outperforms all other methods by a significant margin, from ∼5% to up to ∼20% in different scenarios. In the case of CIFAR-10, which is the most complex of the datasets, we find that FedDiff provides the most improvement. As discussed in our initial motivations, we reason that the iterative nature and focus on sample quality that is provided by the DM objective enables much improved performance for training the global model. Intuitively, this becomes increasingly evident in more complex settings. To verify this observation, we conduct a comparative analysis of generated samples produced by FedCVAE and our FedDiff approach, as illustrated in Figure 2. The discernible disparity is evident, with the samples generated by our method exhibiting significantly enhanced sharpness and overall quality.

## 3.3 NUMBER OF CLIENTS

To further explore the efficacy of our method, we investigate the effect of the number of clients $C$ in Table 2. Note that, as we employ the same total number of samples in all experiments, the number of samples *per client* will increase with smaller $C$, and decrease with larger $C$. This allows us to observe the effect of increasing the distributed nature of the data across a client network.

One question arising from the adoption of generative models in FL settings pertains to their ability to maintain satisfactory performance when trained on a limited number of samples. Interestingly, when analyzing the results, we find that FedDiff is capable of handling a much smaller number of client training samples with little to no performance degradation. On the other hand, the discriminative model approaches quickly experience a collapse in performance when expanding to 20 clients. In the heterogeneous environment of federated learning, the local optimization of a discriminative model on a highly-imbalanced and small dataset proves challenging. Rather than being a burden, such a situation is a blessing for FedDiff, as its sole focus is to capture the subsequently smaller, simpler

distribution. Furthermore, we again find that FedDiff outperforms FedCVAE in all settings, further illustrating the potential for diffusion models in one-shot FL.

Table 2: Results with varying number of clients $C$ with $Dir(0.01)$. As a fixed-size dataset is used in all experiments, increasing the number of clients also decreases the number of samples per client. We find that the SOTA discriminative approaches quickly degrade as the data is distributed across more clients. On the contrary, **our FedDiff maintains strong performance in all settings**.

|  | # Clients | FedAvg | DENSE | OneShot-Ens | FedCVAE | **FedDiff** |
|---|---|---|---|---|---|---|
| FashionMNIST | $C = 5$ | 42.88±3.21 | 47.61±5.44 | 48.91±6.33 | 76.67±2.81 | **86.89±0.34** |
|  | $C = 10$ | 28.62±9.74 | 28.17±17.7 | 33.07±17.5 | 76.31±2.72 | **86.81±0.54** |
|  | $C = 20$ | 27.52±2.98 | 20.10±9.26 | 30.45±8.67 | 76.85±2.57 | **87.24±0.57** |
| PathAMNIST | $C = 5$ | 25.12±3.54 | 30.75±3.01 | 32.48±5.26 | 46.45±1.41 | **72.74±0.63** |
|  | $C = 10$ | 18.21±9.19 | 26.83±10.2 | 30.61±9.55 | 45.16±0.55 | **70.61±1.37** |
|  | $C = 20$ | 15.57±4.13 | 18.11±5.54 | 19.87±6.27 | 41.71±1.89 | **68.96±0.99** |
| CIFAR-10 | $C = 5$ | 23.67±6.12 | 29.58±2.59 | 34.68±3.21 | 29.93±1.70 | **57.68±1.86** |
|  | $C = 10$ | 13.57±3.45 | 20.24±0.84 | 21.49±0.81 | 33.46±2.88 | **56.57±2.42** |
|  | $C = 20$ | 10.02±0.03 | 11.49±2.50 | 11.62±2.71 | 35.13±0.94 | **58.45±0.73** |

## 4 DIFFERENTIAL PRIVACY

Prior one-shot FL works typically have not addressed the aspect of privacy to a satisfactory level. In one-shot FL, where the acquired client model may be repeatedly used or even traded in a model market setting, it is important to investigate different levels of privacy for the model before it departs from the client. Differential privacy is a widely accepted standard for this, as it offers a provable guarantee of privacy. Hence, our analysis includes both discriminative and generative approaches to examine their effectiveness in such scenarios and investigate our research question ***RQ2***.

Table 3: Differential privacy (DP) results under various $\epsilon$ budgets. We set $C = 10$ and $\alpha = 0.01$ as the default setting. **Even under DP constraints, FedDiff is a particularly viable approach, outperforming all other SOTA one-shot FL methods**.

|  | Privacy | FedAvg | DENSE | OneShot-Ens | FedCVAE | **FedDiff** |
|---|---|---|---|---|---|---|
| FashionMNIST | $\epsilon = 50$ | 20.49±12.6 | 31.09±9.87 | 30.41±10.1 | 41.85±2.37 | **75.92±1.86** |
|  | $\epsilon = 25$ | 20.09±12.0 | 29.98±10.5 | 30.70±11.0 | 42.19±2.25 | **75.08±2.13** |
|  | $\epsilon = 10$ | 19.87±12.1 | 29.84±15.2 | 29.29±15.9 | 39.21±2.34 | **73.43±1.50** |
| PathMNIST | $\epsilon = 50$ | 12.93±8.17 | 17.83±6.71 | 20.03±7.19 | 24.31±1.63 | **54.98±2.04** |
|  | $\epsilon = 25$ | 11.18±6.89 | 16.31±3.31 | 19.09±3.56 | 22.65±2.89 | **51.51±1.85** |
|  | $\epsilon = 10$ | 11.21±4.52 | 15.54±1.30 | 17.82±1.92 | 20.97±2.14 | **47.85±3.68** |
| CIFAR-10 | $\epsilon = 50$ | 10.94±0.69 | 14.94±1.98 | 16.01±2.11 | 14.02±0.42 | **32.93±1.93** |
|  | $\epsilon = 25$ | 10.63±0.78 | 14.98±2.37 | 15.84±2.02 | 13.44±1.38 | **31.76±2.68** |
|  | $\epsilon = 10$ | 10.21±1.42 | 15.17±0.78 | 11.34±0.92 | 14.52±2.46 | **27.78±1.66** |

Specifically, we train all approaches under $(\epsilon, \delta)$ differential privacy (DP) at the client level for various privacy levels of $\epsilon = 50$, 25, and 10, with $\delta = 10e^{-5}$, $C = 10$, and $\alpha = 0.01$. Lower $\epsilon$ values correspond to a tighter privacy budget, and the stated budget is for the entire training of each local model. We employ the Opacus (Yousefpour et al., 2021) library for implementing DP. Additional DP training details are provided in Appendix A.1, along with additional $\epsilon$ experiments in Appendix A.2. We present the results for all approaches in Table 3.

As expected, all methods experience a drop in performance when trained under DP settings. Nonetheless, FedDiff still stands out, outperforming all other methods by a significant margin. Particularly for FashionMNIST, FedDiff experiences comparatively less accuracy drop under DP than FedCVAE. As articulated in our initial motivations outlined in Section 3, DP training introduces noise into the training process, exacerbating the complexity of optimization. In such scenarios, the simplicity of the training paradigm employed by diffusion models becomes notably advantageous. Overall, we show that FedDiff is a strong approach even when DP is employed.

## 4.1 ADDRESSING MEMORIZATION

In a recent study, (Carlini et al., 2023) explored diffusion models and identified their ability to memorize samples under certain conditions. They acknowledge differential privacy as the gold standard defense strategy, but did not provide completed experiments to this end. In our investigation, we assess memorization within our DP-trained models to investigate whether inadvertent reproduction of the training data can be eliminated.

To conduct this study, we adopt the evaluation methodology established by (Carlini et al., 2023) to scrutinize the occurrence of memorization. Specifically, from each DP-trained diffusion model, we generate a vast number of samples (five times the size of the training set). Subsequently, for each generated image, we assess potential memorization compared to the original training samples using the adaptive distance metric introduced by (Carlini et al., 2023),

$$\ell\left(\hat{x}, x; S_{\hat{x}}\right) = \frac{\ell_2(\hat{x}, x)}{\alpha \cdot \mathbb{E}_{y \in S_{\hat{x}}}\left[\ell_2(\hat{x}, y)\right]}. \tag{4}$$

Here, $S_{\hat{x}}$ denotes the set comprising the $n$ nearest elements from the training dataset to the example $\hat{x}$. The resulting distance metric yields a small value if the extracted image $x$ exhibits significantly closer proximity to the training image $\hat{x}$ compared to the $n$ closest neighbors of $\hat{x}$ within the training set. The idea is to find generated images that are unusually close an original training image as an indication of memorization. We set $\alpha = 0.5$ and $n = 50$ as in (Carlini et al., 2023).

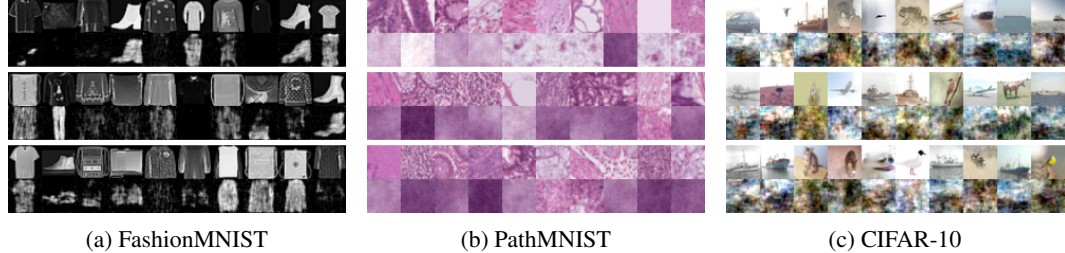

|        (a) FashionMNIST        |        (b) PathMNIST        |        (c) CIFAR-10        |

Figure 3: Qualitative comparison of original training samples and generated samples at $\epsilon = 50$. We show the closest 30 samples via the similarity metric in Equation 4. In each stacked row, the original samples are on top, with the corresponding nearest generated image immediately below. Even under the loosest privacy guarantee of $\epsilon = 50$, we do not see blatant memorization.

(Carlini et al., 2023) did not define the specific threshold for Equation 4 defining when a sample is considered memorized. Therefore, we consider the intuitive threshold to be less than 1, as this would indicate that the distance from the extracted image to the training image is less than half of the average distance to the closest $n$ neighbors. Upon conducting this assessment, we do not find any instances of memorized samples for all datasets under such definition, even at an elevated privacy parameter of $\epsilon = 50$, with the closest distance values being $\sim 1.3$. We show the histogram of scores for all samples on each dataset in Figure 6 of the Appendix.

Because the threshold definition for memorization could vary, we also qualitatively show the samples with the lowest distances for all datasets at $\epsilon = 50$ in Figure 3. Notably, the training versus the generated samples have discernible differences, in contrast to the nearly identical samples uncovered in (Carlini et al., 2023) when training large diffusion models without DP. Also, given the nature of FL, the choice of diffusion model size will typically be small (for example, ours is $\sim 5.8$M parameters), and therefore will be less likely to memorize compared to the larger DMs evaluated in (Carlini et al., 2023). As DP algorithms improve, we anticipate that even better final accuracy can be achieved while maintaining guaranteed privacy in the future with FedDiff.

## 4.2 FOURIER MAGNITUDE FILTERING

While FedDiff performs comparatively well against other SOTA one-shot FL methods under DP constraints, we further investigate a simple approach to improve our method, particularly for complex data most affected by DP. As shown in Figure 3, we note that the generated samples under DP

lack details, exhibiting blurriness and reduced structure. Some images may therefore have notably different frequency spectra compared to the original images, and may actually skew the learning of the global model by muddling classification boundaries and the learned representation.

Therefore, we propose a filtering mechanism to improve the quality of the final synthetic dataset used for global model training. On the client, we take the Fourier transform of the local samples and extract the magnitude information. For each client $c$, we gather the average sample magnitude with

$$\bar{M}_c = \frac{1}{n_c} \sum_{i=1}^{n_c} |\psi(\mathbf{x}^i)|, \tag{5}$$

where $\psi$ is the 2D Fourier transform operation, $\mathbf{x}^i$ is a sample, and $n_c$ is the total number of samples in client $c$. $\bar{M}$ is bundled with the model and transmitted by the client to the relevant global party.

As in our standard global training procedure, samples are generated with the client-trained diffusion models to form a synthetic set of data. Prior to conducting the global training, we calculate a sample score $s$ for the generated data $z$ from each diffusion model from the clients, $s_{\mathbf{z}_c^i} = \||\psi(\mathbf{z}_c^i)| - \bar{M}_c\|_2$.

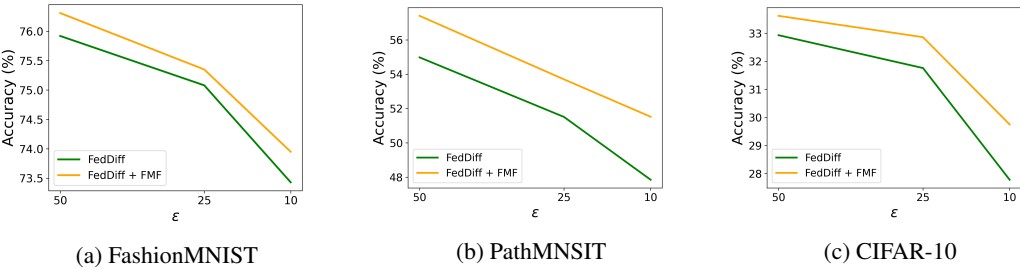

(a) FashionMNIST        (b) PathMNSIT        (c) CIFAR-10

Figure 4: Results with Fourier magnitude filtering (FMF) under DP. FedDiff is in green and FedDiff+FMF in orange. Our FMF approach provides a simple way to boost accuracy, especially in more challenging scenarios such as lower $\epsilon$ budgets and more complex datasets. We plot the mean across three runs with different seeds for each setting. Additional experiments in Appendix A.4.

We can then leverage this information to guide the removal of irrelevant samples. Therefore, The final training set $\mathbf{G}$ is formed by removing $\beta$ percent of the generated data with the highest $s$ (larger magnitude difference). To continually ensure privacy guarantees, we apply DP in the FMF calculation. To do so, we employ the DP bounded mean (Li et al., 2016) from PyDP[1] to calculate the average magnitude $\bar{M}_c$ at each client. This allows us to precisely manage any degree of privacy leakage for $\bar{M}_c$ and include it in the overall privacy budget.

In Figure 4, we show the results of applying our Fourier Magnitude Filtering (FMF) approach with FedDiff for the same overall DP budgets as Table 3. FMF is particularly effective in the most difficult scenarios, helping to mitigate the performance drop in harsh FL environments. For example, FMF provides over 3.5% and 2% improvements with PathMNIST and CIFAR-10 in the challenging $\epsilon = 10$ setting. Overall, FMF is a simple way to boost performance in one-shot FL under DP.

## 5 CONCLUSION

In conclusion, our study addresses two valuable research questions in the context of one-shot FL. Firstly, we investigate the potential of diffusion models in one-shot FL, inspired by their recent prominence in image generation. We explore the specific traits of diffusion models that could offer advantages in this context, and validate our hypothesis through extensive experiments involving our approach, FedDiff, across diverse settings. Secondly, we study one-shot FL methods under provable privacy budgets, and address memorization concerns. Furthermore, to enhance performance under harsh DP conditions, we introduce a simple method, Fourier Magnitude Filtering, to bolster the efficacy of generated data for global model training by eliminating low-quality samples. We hope our work will inspire further research in this direction to improve one-shot FL with generative models.

---

[1]https://github.com/OpenMined/PyDP

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

## A  APPENDIX

### A.1  ADDITIONAL TRAINING DETAILS

The FashionMNIST dataset is an alternative to the original MNIST dataset, providing a more challenging task by replacing the handwritten digits with grayscale images of various fashion items. The dataset consists of 60,000 training images and 10,000 test images. The PathMNIST dataset is a medical dataset of colon pathology images in RGB, with a training set of 89,996 images and a test set containing 7,180 images with 9 classes. The CIFAR-10 dataset consists of 60,000 color images equally distributed into ten different classes. The dataset is composed of a training set containing 50,000 images and a test set comprising 10,000 images. CIFAR-10 is natively sized at $32\times32$ pixels. We upsample FashionMNIST and PathMNIST from $28\times28$ to $32\times32$. A visualization of the dataset partitioning across clients is shown in Figure 5.

We train with a batch size of 128 for all methods and use the AdamW optimizer. For local (and global training were applicable), we searched learning rates from $[3e^{-3}, 1e^{-3}, 3e^{-4}, 1e^{-4}]$ for each method using the CIFAR-10 dataset to find the optimal settings. For DP experiments, we set the max gradient norm clipping threshold to 1.0 for all experiments and methods. In accordance with the recommendations of the Opacus (Yousefpour et al., 2021) library, we employ their Poisson batch sampling to ensure privacy guarantees.

As mentioned in Section 3, our diffusion model is a basic U-Net structure with residual blocks (Ho et al., 2020; Ronneberger et al., 2015) and class-conditioning. For sampling at the server, we perform 1000 iterations as in Ho et al. (2020) to generate each batch. Code will be made available upon acceptance.

### A.2  ADDITIONAL $\epsilon$ EXPERIMENT

To demonstrate the feasibility of FedDiff under more stringent budget constraints, we conduct an experiment with an even tighter privacy budget of $\epsilon = 1$ in Table 4. Despite facing such stringent privacy constraints, FedDiff maintains a higher level of performance at $\epsilon = 1$ than all other methods in Table 3 at $\epsilon = 50$.

Table 4: Differential privacy (DP) results under $\epsilon = 1$. Even in this case, FedDiff outperforms all other methods with the much larger budget of $\epsilon = 50$ in Table 3.

| Privacy $\epsilon = 1$ | **FedDiff** |
|---|---|
| FashionMNIST | **65.53±0.70** |
| PathMNIST | **44.38±3.35** |
| CIFAR-10 | **21.48±1.53** |

### A.3  CLIENT EXPERIMENTS

We conduct further experiments using a ResNet16 (He et al., 2015) architecture with approximately 6.4M parameters as the client models for FedAvg, DENSE, and OneShot-Ens, as well

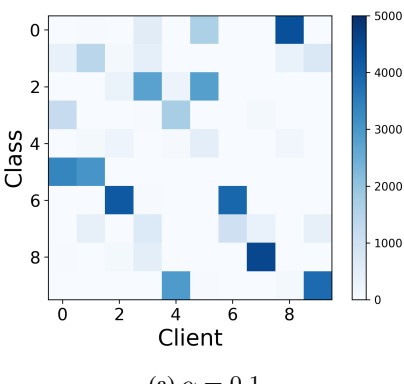 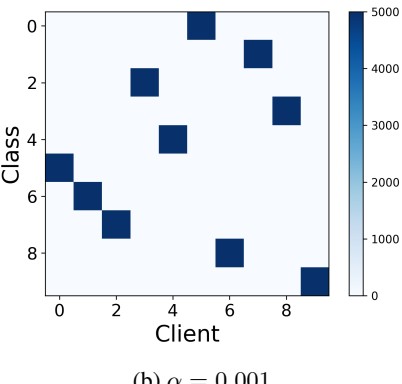

(a) $\alpha = 0.1$           (b) $\alpha = 0.001$

Figure 5: $Dir(\alpha)$ data partitioning for 10 clients on CIFAR-10. We show moderate ($\alpha = 0.1$) to severe ($\alpha = 0.01$) data heterogeneity levels. Data heterogeneity poses a significant challenge for many one-shot FL methods, as reconciling various models trained on widely different distributions is non-trivial. Our FedDiff approach rather trains diffusion models on the simple client distributions, which can then generate useful synthetic data for training global models.

as the server model for DENSE, FedCVAE, and FedDiff. This ensures that all discriminative (local and global) and generative models are of similar number of parameters and resource requirements, rather than the simpler network typically used in (Zhang et al., 2022; Heinbaugh et al., 2023) and described in our main paper. In Table 5, we see that using the ResNet architecture for local and global models improves performance for all methods in comparison to the same standard setting ($C = 10$, $\alpha = 0.01$) in Table 1 of the main paper. Notably, FedDiff actually benefits the most from the improved global model architecture, illustrating that the generated data from FedDiff can be leveraged effectively with different global architectures.

Table 5: Results with ResNet16 with our standard FL setting ($C = 10$, $\alpha = 0.01$).

|  | FedAvg | DENSE | OneShot-Ens | FedCVAE | **FedDiff** |
|---|---|---|---|---|---|
| CIFAR-10 | 18.93±2.98 | 20.96±2.69 | 22.93±4.34 | 35.67±2.19 | **59.94±2.01** |

Furthermore, we take note of the resource factors, including FLOPs and parameter count, for each method on a single client. Notably, our FedDiff approach consistently delivers superior accuracy while requiring comparable computational resources to other methods. We extend this assessment to a reduced model size (FedDiff$_s$ in Table 6), reaffirming its strong performance relative to alternative methods. This analysis underscores the effectiveness of FedDiff, even when deployed on hardware with modest computational capabilities.

Table 6: Accuracy versus FLOPs and parameter count (Params) for each method on a single client. Our FedDiff approach consistently attains heightened accuracy levels while maintaining very reasonable resource demands on par with other methodologies. We also evaluate our method with a scaled-down model variant (FedDiff$_s$), confirming its performance relative to alternative approaches. This analysis underscores realistic feasibility of FedDiff framework.

| Metrics | FedAvg | DENSE | OneShot-Ens | FedCVAE | **FedDiff** | **FedDiff$_s$** |
|---|---|---|---|---|---|---|
| MFLOPs ↓ | 479.92 | 479.92 | 479.92 | 79.00 | 301.14 | **77.43** |
| Params ↓ | 6.44M | 6.44M | 6.44M | 5.97M | 5.81M | **1.46M** |
| Acc. (%) ↑ | 18.93±2.98 | 20.96±2.69 | 22.93±4.34 | 35.67±2.19 | **59.94±2.01** | **48.56±2.89** |

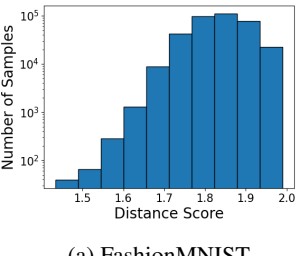 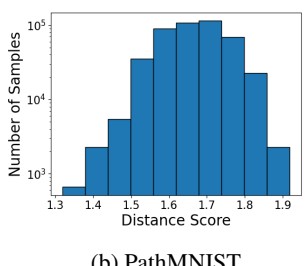 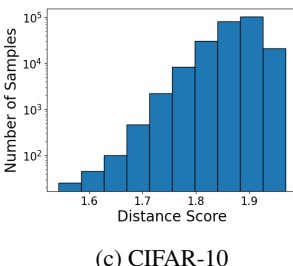

(a) FashionMNIST  (b) PathMNIST  (c) CIFAR-10

Figure 6: Histogram of distance scores for all generated samples to corresponding closest training image by Eq. 4 on each dataset. Note that the y-axis in in *log scale*, as there are very few samples with lower scores.

## A.4 FMF $\beta$ ABLATION

In Figure 7, we present the outcomes obtained using FedDiff+FMF under $\epsilon = 10$ across a range of $\beta$ values, encompassing data filtering percentages spanning from 1% to 12%. Our findings indicate that, in general, data filtering within the 1% to 10% range yields favorable results and leads to performance enhancements, with around 5% being a great default. Interestingly, the the degree of improvement provided by FMF becomes more pronounced and consistent as the dataset becomes more challenging. This phenomenon aligns with the anticipated trends, as more intricate datasets inherently pose a greater challenge, making it less likely for the generators to consistently produce high-quality samples. Consequently, the need for data filtering becomes more pronounced in such scenarios to enhance sample quality. This trend is also favorable since it addresses the specific need for improvement, especially in cases where performance is suboptimal and the challenges are more pronounced.

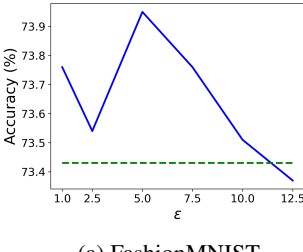 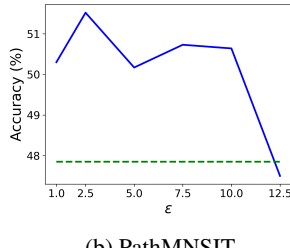 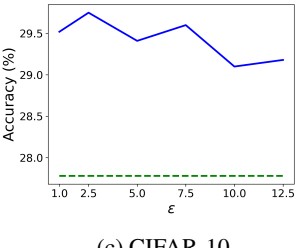

(a) FashionMNIST  (b) PathMNSIT  (c) CIFAR-10

Figure 7: Ablation study of $\beta$ in FMF under the $\epsilon = 10$ setting. The accuracy of FedDiff is in green and FedDiff+FMF for various $\beta$ in blue. Generally, data filtering within the range of 1% to 10% produces positive outcomes, resulting in improved performance, with approximately 5% serving as an effective default choice. We plot the mean across three runs with different seeds for each setting.

## A.5 DISCUSSIONS, LIMITATIONS AND BROADER IMPACT

**Model Heterogeneity**. In real FL systems, model heterogeneity may often occur (Zhang et al., 2022; Heinbaugh et al., 2023). For instance, some clients may have architecture variations in their models or have smaller or larger models depending on their computing capabilities. Therefore, clients may have different architectures of similar generation capability, or even differing capabilities depending on the requirements of each client system. Our approach allows for flexibility to accommodate such system diversity across clients. In FedDiff, we generate data from the client models and employ that synthetic data for global training, and therefore can leverage varying models without the worry of reconciling the weights themselves.

**Limitations and Broader Impact**. One downside of our method is that the generated data, particularly under DP constraints, still lacks in quality and effectiveness for global model training versus

using true data. For instance, with DP on CIFAR-10 as shown in Figure 3, the data loses a substantial amount of structure. An interesting direction for future work would be to study how to improve the quality of the generated data and its usefulness for global model training while maintaining privacy.

Looking at the broader impact of our work, FL depends on the diversity of data contributed by different participants. If biases exist in the local datasets, they can be propagated and amplified during the model training process. This could lead to unintended algorithmic biases and discrimination in the resulting models. Ensuring diversity and fairness in the data used for FL is an important research direction to mitigate this risk and promote equitable outcomes (Abay et al., 2020), particularly in the highly data heterogeneous environments explored in this work. Furthermore, as we have discussed throughout our paper, the privacy of client data is important in FL. To mitigate risks in this regard, we take many precautions to preserve privacy of the clients participated in the FL process though the use of DP, and operating within the one-shot setting to reduce the chance of eavesdropping.

