# OpenReview forum: "Exploring the Effectiveness of Diffusion Models in One-Shot Federated Learning"
_ICLR.cc/2024/Conference — ICLR 2024 Conference Withdrawn Submission_

### Official Review · Reviewer_TnbG · 2023-10-31

**Soundness:** 3 good
**Presentation:** 3 good
**Contribution:** 2 fair
**Rating:** 6
**Confidence:** 4

**Summary:**

In this paper, the authors proposed to utilize diffusion model in one-shot federated learning for image domain application. In particular, each party trains a diffusion model using their own local private dataset and shares the trained diffusion models with the aggregator, and then the aggregator uses the shared diffusion models to generate synthetic data samples to train a global model. The authors also proposed to apply differential privacy (DP) to train the local diffusion models such that the shared local diffusion models will not reveal parties’ private training data under DP guarantee. Finally, they suggested to use Fourier Magnitude Filtering (FMF) to filter out low quality generated samples before using them in global model training. Based on their empirical experiments, the proposed algorithm FedDiff can outperform several existing baselines for one-shot federated learning on multiple datasets especially in highly heterogeneous settings.

**Strengths:**

Overall, the paper is easy to understand, and the proposed algorithm seems novel. In the numerical experiments, the authors tested the algorithm on several datasets, different level of data heterogeneity and sample size.

**Weaknesses:**

I can see a few weaknesses, and listed as follows:
1. The proposed algorithm can only be applied to image domain, but in general FL can be applied to other domains, such as text and tabular data.
2. The authors only empirically verify the proposed method and provided very little theoretical insights on why diffusion model is a better choice than other existing one-shot FL algorithms.

**Questions:**

1. It is unclear if diffusion models can reveal parties’ private training data under label-wise data heterogeneity case. Under this scenario, the aggregator can tell if a party has samples belonging to certain class or not.
2. There are two related work on one-shot federated learning the authors failed to compare with.
> Yurochkin, M., Agarwal, M., Ghosh, S.S., Greenewald, K.H., Hoang, T.N., & Khazaeni, Y. (2019). Bayesian Nonparametric Federated Learning of Neural Networks. ArXiv, abs/1905.12022.

> Witherspoon, S., Steuer, D., Bent, G.A., & Desai, N.V. (2020). SEEC: Semantic Vector Federation across Edge Computing Environments. ArXiv, abs/2008.13298.

The first paper seems to achieve better performance on CIFAR10 dataset, and the second paper can be applied to text datasets.

3. When the number of parties $C$ reaches 20, the per client sample size is still large for many of these datasets. However, in LEAF benchmark, for FEMNIST, many clients only have around 100 samples. I wonder if diffusion models can still work in those cases.
4. The authors did not discuss whether it is beneficial for the aggregator to aggregate the local diffusion models and used the aggregated model to generate better synthetic samples. It would be good to compare the proposed algorithm with the above idea.
5. Why DP affects FedDiff’s performance on CIFAR10 data the most?

---

### Official Review · Reviewer_dW3r · 2023-11-01

**Soundness:** 3 good
**Presentation:** 2 fair
**Contribution:** 2 fair
**Rating:** 5
**Confidence:** 3

**Summary:**

This work explores the effectiveness of diffusion models in one-shot FL, demonstrating their applicability in addressing data heterogeneity and improving FL performance.

**Strengths:**

This paper has a good level of writing and it is easy to follow. The idea is easy to follow and understand.

**Weaknesses:**

1. The methodology is not novel. It replaced GAN or CVAEs with the diffusion model.
2. Some important issues are not discussed in the paper, e.g., the cost of training a diffusion mode.

**Questions:**

1. Diffusion model is a time-consuming model. Can the author provide more detailed experiments about the cost of training the models in the clients?

2. Some important ablation studies are missing, e.g., timestep, sampling steps, and sampling variances. Can the author provide some results about this?

---

### Official Review · Reviewer_fpTS · 2023-11-02

**Soundness:** 2 fair
**Presentation:** 2 fair
**Contribution:** 2 fair
**Rating:** 3
**Confidence:** 4

**Summary:**

The paper proposes to use diffusion models in one-shot federated learning scenario. Every client trains a local diffusion model locally and sends the corresponding diffusion model to the server. The sever collects all the diffusion models to generate the synthetic data to train a final global model. To tackle the privacy issue, differential privacy mechanism is adopted locally in the training of local diffusion models.

**Strengths:**

The paper is not hard to follow. The motivation of the work is clear.

**Weaknesses:**

1)	The methodology proposed is a simple introduction of (differential private) diffusion model in one-shot FL with generative models. The only novelty part might be the Fourier Magnitude Filtering (FMF) scheme introduced in the end of paper, but without much exploration (both in experimental and theoretical perspectives).
2)	The procedure of how many data points is generated from each diffusion model is not clearly indicated in the procedure. This is important since it will impact the final synthetic data distribution and thus the global model performance. Besides, how this procedure should be designed with consideration of the proposed FMF needs to be further investigated.
3)	As stated in the paper, the interest of using diffusion model other than CVAE in one-shot FL is the better sample quality of generated data. However, on the other hand, CVAE normally has better diversity than diffusion model. The experimental settings considered in the paper are only the extreme heterogenous case (alpha<=0.1) that every client has a really small diversity in classes/samples.  One would expect FedCVAE performs better and FedDiff performs worse in a less heterogenous scenario. However, the paper did not explore this setting.
4)	The paper needs to be reorganized where the algorithm is needed to be detailed with FMF and DP. Besides, the appendix A.3 needs to be better integrated into the main paper, as for now it seems to be a response section for rebuttal. If FedDiff works well for a more realistic model like ResNet-16 studied in A.3, as well as a more private scenario $\epsilon=1$, it is better to have a complete result on this model with more realistic privacy regime in the main paper other than small CNN’s ones presented.

**Questions:**

1)	From Table 2, DENSE on FashionMNIST for C = 10 gives 28.17%. However, in Table 3, DENSE under DP returns better accuracy which contradicts the conclusion made “all methods experience a drop in performance when trained under DP settings”.
2)	From what we observed in Table 1 and 2, for dataset PathMNIST, FedDiff’s performance in terms of accuracy decreases as the client’s dataset becomes more imbalanced in classes, and the dataset becomes smaller as well. Shouldn’t FedDiff’s performs better as the diffusion model captures better the local data distribution as stated in Section 3.3?

Typos:
1) Conclusion: “to bolster” -> “to booster”
2) Figure 1 caption: two $z_0$
3) Figure 7, the x-axis should be $\beta$ instead of $\epsilon$.

---

### Official Review · Reviewer_9S34 · 2023-11-03

**Soundness:** 2 fair
**Presentation:** 3 good
**Contribution:** 3 good
**Rating:** 5
**Confidence:** 4

**Summary:**

The paper provided an overview from an empirical aspect of incorporating diffusion models as the data generators in one-shot Federated learning. Experiments have shown that the above approach provides observable accuracy gain compared to the baseline methods.

**Strengths:**

1. The paper not only conducted experiments on the accuracy aspect but also evaluated other benchmarks like differential privacy and memorization effect.

2. The detailed settings of the experiments are provided in the paper.

3. The algorithm is evaluated on several real-world datasets to demonstrate the effectiveness of the proposed approach for FL of diffusion models.

**Weaknesses:**

1. Diffusion models are notoriously known to be computationally expensive regarding training. However, in the usual setting of federated learning, there always exists device heterogeneity, and a lot of less capable devices in the scheme may cause huge overheads to the whole training process if they are forced to train a diffusion model locally or even straight up not being able to do so. This problem needs further investigation.

2. All datasets used in the experiments consist of low-resolution pictures. It would be nice to see the results on large-scale datasets like ImageNet.

3. There needs to be a theoretical or hypothetical analysis of some of the discovered results in the paper. Why the well-observed memorization effect did not happen? What could be the theoretical rationale behind this? These problems remain to be addressed.

**Questions:**

Weaknesses